



# Comparing Flood Forecasting and Early Warning Systems in Transboundary River Basins

Tim Busker[1], Daniela Rodriguez Castro[2], Sergiy Vorogushyn[3], Jaap Kwadijk[5,8], Davide Zoccatelli[7], Rafaella Loureiro[1], Heather J. Murdock[4], Laurent Pfister[7], Benjamin Dewals[2] , Kymo Slager[5], Annegret H. Thieken[4], Jan Verkade[5], Patrick Willems[6], and Jeroen C.J.H. Aerts[1,5]

[1] Institute for Environmental Studies (IVM), VU University Amsterdam, Amsterdam, The Netherlands.
[2] Research Group of Hydraulics in Environmental and Civil Engineering (HECE), University of Liège, Liège, Belgium.
[3] GFZ Helmholtz Centre for Geosciences, Potsdam, Germany.
[4] Institute of Environmental Science and Geography, University of Potsdam, Potsdam, Germany.
[5] Deltares, Delft, The Netherlands.
[6] Hydraulics and Geotechnics Section, KU Leuven, Leuven, Belgium.
[7] Luxembourg Institute of Science and Technology (LIST), Esch-sur-Alzette, Luxembourg.
[8] Faculty of Engineering Technology, University of Twente, Enschede, The Netherlands.

*Correspondence to*: Tim Busker (tim.busker@vu.nl)

**Abstract.** This study compares operational Flood Forecasting and Early Warning Systems (FFEWSs) in transboundary river basins in Northwestern Europe, covering parts of Luxembourg, Germany, The Netherlands and Belgium. This region was hit by an extreme flood event in 2021 with over 200 fatalities. Due to the high death toll, FFEWSs were heavily criticized in the aftermath. Expert interviews from the region revealed strong improvements of the FFEWSs after this flood event in all countries. All regions have invested in probabilistic flood forecasting systems, and all countries now use mobile phone-based alerts. Strong differences in flood warning levels and color codes exist across and within the countries. In response to the 2021 flood, some regions have introduced an additional purple warning level. The interviews also revealed that the uptake of operational impact-based forecasts remains challenging, while these are crucial for translating hydrological forecasts to effective actions. For example, interviewees highlighted the need for operational flood inundation forecasts. However, Flanders is the only region where such forecasts are provided. It is recommended to enhance forecasts with impact-based information, including inundation maps delineating the people and objects at risk. This can improve the early actions taken by first responders and the affected people.





## 1 Introduction

The extreme flood event of July 2021 in North-Western Europe caused over 200 fatalities and around €40bn of damage (Lehmkuhl, et al., 2022). The frequency and severity of such flood events may increase in the future because of climate change and socio-economic developments (Tradowsky et al., 2023). To reduce avoidable damage and casualties, there is a call for more reliable early warning systems to protect Europe from flood impacts (European Environment Agency, 2024). Forecast-based actions are vital to reduce disaster risk, as they can effectively reduce the exposure and vulnerability of communities to

floods (Pappenberger et al., 2015). For example, based on forecast information, the government may issue an evacuation order for communities in predefined flood zones or may advice households and businesses to protect properties and infrastructure with emergency flood protection measures (e.g. sandbags and pumps). The benefits of forecast-based actions are often much higher than their costs (Global Commission on Adaptation, 2019). Pappenberger et al. (2015) even claim a benefit in the order of 400 Euro for every Euro invested. The 2021 flood further showcased the potential of early warning systems. Extreme

precipitation was predicted by weather models, such as the model operated by the German Weather Service (DWD, 2021; Mohr et al., 2023), which issued extreme weather warnings 1-2 days in advance (DWD, 2021; KNMI, 2021). Despite these promising numbers, the impacts of the flood event were devastating and raised questions about the functioning of the operational flood warnings and the effectiveness of early actions taken.

A flood forecasting and early warning system (FFEWS) consists of a chain of components (Fig. 1). Hydro-meteorological data is used as input to weather and flood forecasting models, which jointly predict whether warning thresholds are surpassed with a certain probability (e.g., Alfieri et al., 2019). These thresholds can be, for example, pre-defined rainfall amounts or water levels with a certain return period. Surpassing a threshold may lead to a warning to the public or crisis managers who make decisions on how to respond (Fig. 1). Meteorological and hydrological warning levels are often depicted with colors, as shown

in Fig. 1. Each warning level can be connected to a set of pre-defined actions, outlined in crisis management plans. Responses at the governmental level include, for example, closing a certain parking garage because it might be flooded, or in severe cases issuing an evacuation alert or order. Households may decide to install flood protection measures such as sandbags or mobile protection systems (Cao et al., 2024).

Uncertainty is present in each component of the FFEWS. In the forecasting part, uncertainty can be represented by using ensemble weather forecasts (Fig. 1, left). In addition, uncertainty of forecasts increases with lead time (i.e., the time between issuing a forecast and the real event) - usually one or a few days ahead of the event (Jiang et al., 2023). Further uncertainties in the system lie in the communication, decision-making, and response stages, and whether decision-makers can process warning signals into effective actions on the ground (Bischiniotis et al., 2020). A key challenge in developing and

implementing a FFEWS is that uncertainty in one component can cascade down to the next component, all the way to the last component (Parker and Priest, 2012). For example, uncertainty in the collection and processing of meteorological data (Fig.



1, Left) may lead to a 'missed' forecast where thresholds in the system were not surpassed, while observed water levels reached extreme heights (Cloke and Pappenberger, 2009). By contrast, uncertainties can also lead to so-called false alarms, where the predictions suggest that warning thresholds may be exceeded, while effectively are not.

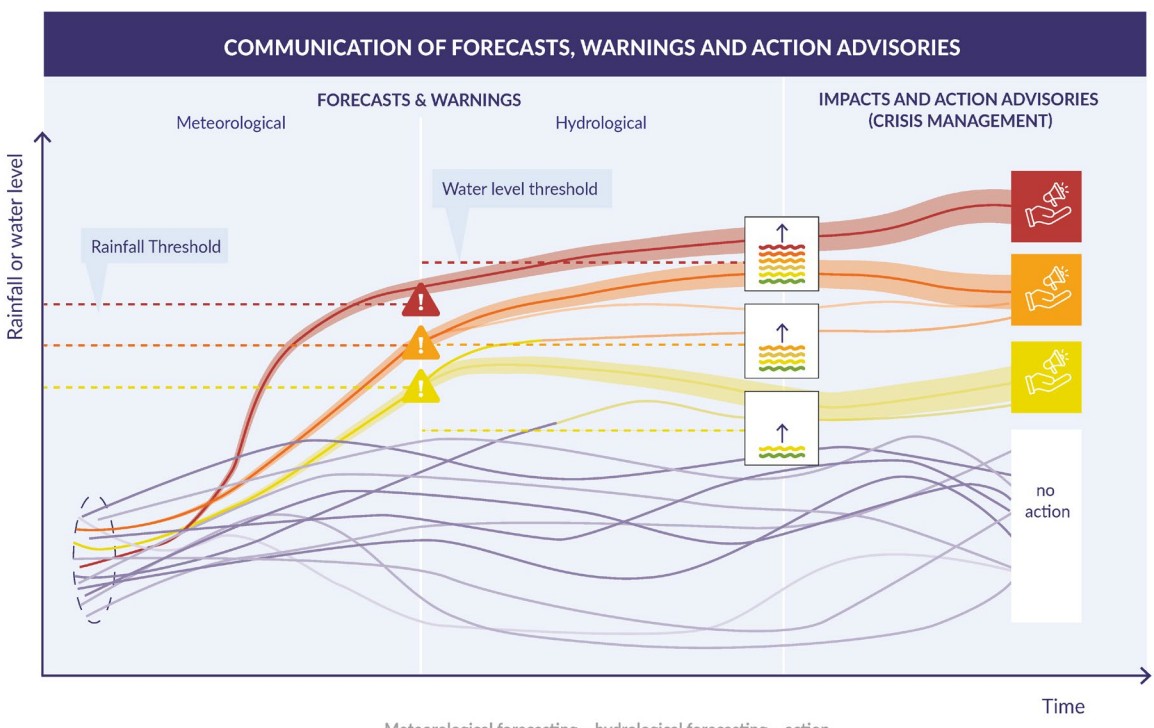

**Figure 1: Conceptual representation of a flood forecasting and early warning system (FFEWS). Rainfall ensemble forecasts (*left*) are used in hydrological models to predict river water levels (*middle*). Warnings are issued if a share of the ensembles exceed the specified rainfall or water level threshold, which results in triggering (sometimes pre-determined) early actions (*right*) (adapted from: Busker, 2024).**

The quality of meteorological forecasts is challenged by the inherent uncertainties in atmospheric processes and the related inaccuracies in both spatio-temporal estimations of initial conditions and forecast simulations (e.g., Leutbecher and Palmer, 2008). The first source of error relates to input data for forecasting models, while the second arises from the imperfect physical parametrization and (coarse) resolutions of models (Buizza, 2019). Numerical weather prediction models have improved greatly in recent decades (Bauer et al., 2015). For example, the Integrated Forecasting System (IFS) operated by the European Centre for Medium-Range Weather Forecasts (ECMWF) produces a 51-member ensemble forecast, which can be used to derive probabilities. Applying these data to derive information needed for warning systems requires the definition of a threshold, for example rainfall greater than 20 mm/h. The predicted probability of such a precipitation event could be defined as the percentage of ensemble members exceeding this threshold.





As with meteorological forecasts, the quality of the hydrological forecasts is affected by various factors, such as meteorological
observations or hydrological characteristics (e.g. ground- and soil water volume). Over the last decade, ensemble and
probabilistic techniques have been increasingly used to estimate the uncertainty in hydrological forecasts. Despite advances,
early warning systems only have value if they lead to effective early actions (Golding et al., 2019). Here, the communication
of flood warnings to first responders and the public appears challenging (Dasgupta et al., 2024; Parker and Priest, 2012;
Scolobig et al., 2022). For example, Thieken et al. (2023b) showed that around a third of the flood-related fatalities in the
German region North Rhine-Westphalia (NRW) were likely not warned, despite having early-warning systems in place. In
addition, from the people that were warned, 85% did not expect severe flooding (Thieken et al. 2023a). In the Netherlands,
75% of people with flooded homes in the heavily affected Geul catchment were not warned (Endendijk et al., 2023a). In the
valley of the Vesdre river in the Walloon region of Belgium, households were unexpectedly hit by the flood as the flood event
was much larger than the official flood zone delineated by governmental zoning policies (Dewals et al., 2021). In Germany,
50 to 75% of flood fatalities happened outside of the marked flood hazard zones in NRW and Rhineland-Palatinate (RLP),
respectively (Rhein and Kreibich, 2025; Thieken et al. 2023b). Flood risk awareness and risk perception are important drivers
of adaptation: since households not living in flood zones have lower risk perceptions, their responses are also lower than those
living in flood zones (Aerts et al., 2018). Empirical data on flood zone residents in The Netherlands show that households who
received a flood warning were 23% up to twice as likely to take emergency flood risk reduction measures as those who did
not receive a warning (Endendijk et al., 2023a).

These findings underline the importance of the provision of timely warnings that include predicted impacts (Najafi et al., 2024)
and actionable information (Kreibich et al., 2021). Impact-based forecasts are more specific and tangible (Merz et al., 2020),
trigger change in people's risk perceptions, and consequently their intention to respond (Endendijk et al. 2023b; Red Cross
Red Crescent Climate Centre, 2020). This points to the importance of clear communication of the expected flood, the expected
impacts, and of providing recommendations for responses (e.g., Thieken et al., 2023a). In addition, many countries have not
yet homogenized their FFEWS protocols, which can potentially hamper clear and consistent cross-border communication
(Coughlan de Perez et al., 2022). This is caused by, for example, differences in institutional settings and geographical
characteristics in the respective countries. The Council of the European Union states that a better integration of activities and
cooperation between countries in cross-border river-basins is crucial to further develop FFEWSs (Council of the European
Union, 2024).

Given these challenges, the main goal of this study is to compare FFEWSs in transboundary river basins in Northwestern
Europe hit by the July 2021 flood. We intend to gain and share knowledge on FFEWSs applications in a transboundary context
and to develop recommendations for their further improvement. We focus on two questions related to the use of FFEWSs: (a)
what are the differences in the warning levels across countries and how are they defined?, and (b) how can the communication



of early warning information and the translation into effective action be improved? We collected our data through semi-structured interviews and a literature review about FFEWSs in The Netherlands, Germany, Belgium and Luxembourg.

## 2 Case study region and approach

### 2.1 Case study region

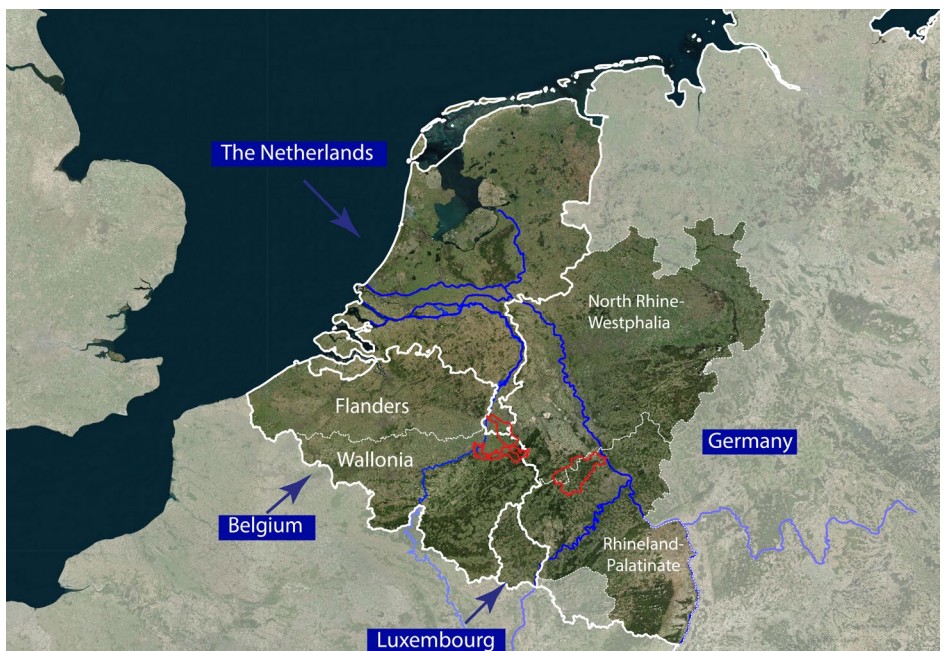

**Figure 2: The study area in Northwestern Europe with the transboundary Rhine and Meuse rivers, shared by The Netherlands, Germany, Luxemburg and Belgium. These countries and sub-regions were severely impacted by the extreme flood event in July 2021, especially in smaller transboundary rivers such as the Geul, Vesdre and Ahr (red polygons).**

The flood event of summer 2021 severely hit the area shown in Fig. 2, which compromises several countries that share the

Meuse and Rhine basins. The event was caused by an atmospheric low-pressure system delivering a total precipitation amount of about 200 mm in the Ardennes-Eifel region over 48 hours (Lehmkuhl et al., 2022, Mohr et al., 2023). The impacts were the largest in the valleys of smaller tributaries such as the Ahr and Vesdre (red polygons), with the most severe damage in the Ahr valley, where 135 people lost their lives (out of total 190 fatalities in Germany; see Thieken et al. 2023b). In Belgium, parts of the Vesdre valley were devastated, where in total 39 people lost their lives (Journée et al., 2023). For The Netherlands, no

fatalities were reported, but damage amounted to more than half a billion euro –particularly in the Geul tributary (ENW, 2021). Also in Luxembourg, river gauges reported unprecedented water levels (> $HQ_{100}$) which led to widespread inundation and evacuation orders (AGE, 2021).





*Flood risk management*

Flood risk management is organized quite differently in the riparian countries of the Meuse-Rhine area. In The Netherlands, the national government is managing the large waterways (e.g., Rhine and Meuse), while regional governments (including water boards) are responsible for the smaller regional water systems. In Germany, flood risk management is the responsibility of each of the sixteen federal States (e.g., Becker et al., 2015). Hence, we particularly focus here on flood forecasting and warning deployed in the riparian states of RLP and NRW in Germany. In Belgium, the two federal states (Flanders and Wallonia) both have the mandate to develop and enforce flood risk management regulations. In Luxembourg, flood risk management is addressed at the national level by the Water Management Administration (AGE), which is part of the Ministry of the Environment, Climate and Biodiversity.

**2.2 Approach**

We collected information on FFEWSs in the riparian regions from literature reviews and interviews with 13 experts in the countries of the case study area. We applied a literature search using (combinations of) keywords such as #flood, #forecast, #threshold, #communication, #response. We also manually scanned country-specific reports that describe or evaluate flood forecasting systems in national languages. Interviews were held with 13 flood forecasting experts from national hydro-meteorological agencies or with agencies and experts involved in crisis management (Appendix A). For the interviews, we developed a semi-structured questionnaire (Supplement S1) to inquire about specific questions related to evaluating current FFEWSs. We structured the interview in the five different FFEWS pillars, as outlined by WMO (2023): (1) data collection, (2) forecasting, (3) communication, (4) decision making, (5) action/response. The division of the interview into five pillars facilitated a thematic analysis (Kiger & Varpio, 2020). Within each pillar, we synthesized the responses, identifying patterns, commonalities, and differences based on the audio recordings obtained with the participant's consent. We included an additional iteration of the synthesized material with most interviewees. Below, we discuss the results for each country. To contextualize our findings, we integrated insights from the literature, comparing them with state-of-the-art scientific research.

**3 Key characteristics of the FFEWS**

Figure 3 and Table 1 provide an overview of different characteristics of the FFEWSs in the regions and countries hit by the 2021 flood event. Figure 3 shows that flood warning levels differ between countries, as well as between regions. For example, Flanders and Wallonia, or NRW and RLP, have different color codes, which are defined in a different way.

With regards to meteorological warnings, the meteorological agencies generally use 3 rainfall thresholds, except for Luxembourg where 4 levels are used. In the Netherlands and Belgium, yellow-orange-red colors are used as warning levels. Luxembourg and Germany recently added an extra warning level (dark purple) to represent events with immediate danger. For the lowest threshold level, all countries use thresholds of ~30mm/hr or ~40mm/6hrs, whereas the highest threshold is often





related to an event of 80 to 100mm/24hrs (among other thresholds). Only in The Netherlands, the red warning level is not distinguished from orange based on rainfall but only based on expected impacts. These impacts are estimated in an impact-team meeting with key stakeholders (see Section 3.3).

The number of hydrological warning levels varies between 3 to 6 (Fig. 3). Hydrological thresholds are mostly clearly defined in all regions, except for Wallonia. Here, solely expert judgement is used to determine the warning level. Red is the highest warning level, except for RLP region where a dark purple level was added to represent catastrophic floods (with a 100-year return period and beyond). The thresholds are based on either return periods (e.g., in RLP), or local flood impacts (e.g., in NRW, Luxembourg, Flanders). Warnings in Flanders slightly deviate from the other countries and use different water level

thresholds for: (a) Alarm States, based on impact to water infrastructures, and (b) Flood Warnings, based on the distance between the water level and dike crest (Fig. 3 and Section 3.4).

Table 1 shows that most FFEWSs follow a probabilistic approach to account for the uncertainty in the forecasting system. Where the other regions rely on discharge forecasts and thresholds, Flanders is the only region where flood inundation forecasts

run operationally (for their short-term forecasts, 48h). The three columns on the right of Table 1 show information related to responses. All countries except Wallonia and Luxemburg have mandatory flood emergency plans. Meanwhile, all regions also have a cell phone-based alert system and have online portals with flood forecast information, which are updated regularly during flooding situations. In Germany and Luxembourg, a cell broadcasting system was installed in response to and after the July 2021 flood.


We now proceed with an overview per country with (a) a general description of the FFEWSs and a more in-depth assessment of the current (b) meteorological and (c) hydrological forecasts, and (d) discussion on the characteristics of crisis management and communication, with an emphasis on experiences during the 2021 flood event.




**Figure 3: An overview of the operational early-warning levels for pluvial (left) and fluvial (right) floods. The warning colors correspond to the colors used operationally.**

| WARNING LEVELS AND THRESHOLDS | | |
|---|---|---|
| Country | Pluvial floods (National Meteorological Services) | Fluvial floods (Hydrological services) |
| The Netherlands | ⚠️ ≥ = 30 mm in 1 h (thunderstorms) or ≥ 50 mm in 24 h<br><br>⚠️ ≥ 50 mm in 1 h or ≥ 75 mm in 24 h (heavy rain)<br><br>⚠️ On the decision of the Weather Impact Team | *Large rivers: based on impact-adjusted return periods*  *Regional rivers (water boards): thresholds depending on the water board*<br><br>1/25 to 1/50 years / 1/5 to 1/10 years / Yearly / sub-yearly |
| Belgium | ⚠️ 20 to 30 mm in 1 h or 20 to 40 mm in 6 h or 25 to 50 mm in 24 h<br><br>⚠️ 31 to 50 mm in 1 h or 41 to 60 mm in 6h or 51 to 100 mm in 24 h<br><br>⚠️ Already flooding problems and heavy rain still forecasted or 50 mm in 1 h or 60 mm in 6h or 100 mm in 24h | *Flanders:* *Alarm states: water level thresholds, based on impact to water infrastructures* *Flood warnings: water level thresholds, based on distance between water level and dike crest*   *Wallonia:* *Tresholds not present, warning level selected based on expert-judgement* |
| Luxembourg | ⚠️ Awareness level yellow / Be aware (low risk / potential danger) 15-30 mm in 6h or 20-40 mm in 12h or 30-50 mm in 24h<br><br>⚠️ Awareness level orange / Be careful (medium risk / danger) 31-45 mm in 6h or 41-60 mm in 12h or 51-81 mm in 24h<br><br>⚠️ Awareness level red / Utmost vigilance (high risk / significant danger) >45 mm in 6h or >60 mm in 12h or >80 mm in 25h<br><br>⚠️ Imminent danger / Immediate action (flash floods) | *Based on local flood impact* |
| Germany | ⚠️ 15 to 25 mm in 1 hour or 20 to 35 mm in 6 h or 25 to 40 mm in 12 h or 30 to 50 mm in 24 h or 40 to 60 mm in 48 h or 60 to 90 mm in 72 h<br><br>⚠️ 25 to 40 mm in 1 hour or 35 to 60 mm in 6 hours or 40-70 mm in 12 h or 50-80 mm in 24 h or 60-90 mm in 48 h or 90-120 mm in 72 h<br><br>⚠️ > 40 mm in 1 hour or > 60 mm in 6 h or > 70 mm in 12 h or > 80 mm in 24 h or > 90 mm in 48 h or > 120 mm in 72 h | *North Rhine Westphalia* *Based on local flood impact*  *Rhineland-Palatinate* *Based on return periods*<br><br>≥ 100 year / ≥ 50 year / ≥ 20 year / ≥ 10 year / ≥ 2 year / < 2 year |





**Table 1. Characteristics of Flood Forecast and Early Warning Systems (FFEWS) in different European regions and countries.**

| Country | Deterministic or probabilistic hydrological forecast | Type of hydrological forecast (D, I) | Lead time (h) | Flood emergency plans (Y/N) | Primary alerting system | Flood forecast online portal |
|---|---|---|---|---|---|---|
| Germany (RLP) | PF | D | 48 hours | Y | MoWas and DE-alert | National: www.hochwasserzentralen.de https://www.hochwasser.rlp.de/ |
| Germany (NRW) | PF | D | 10 days | Y | | https://hochwasserportal.nrw/lanuv (Only observations) |
| Luxembourg | PF | D | 24-48h | Y | LU-alert | https://www.inondations.lu/map |
| The Netherlands | DF (5-day forecast) PF (15-day forecast) | D | 5- days and 15 days | Y | NL-alert | https://waterinfo.rws.nl/#/ waterberichtgeving.rws.nl |
| Belgium (Flanders) | PF | D (10-day forecast) I (48h-forecast) | 48h and 10-days | Y | BE-alert | https://www.waterinfo.vlaanderen.be/ |
| Belgium (Wallonia) | PF | D | Depending on cathment size | Y but not for all areas | | https://hydrometrie.wallonie.be/home.html |

D= Discharge        PF=probabilistic forecast
I = Inundation        Y=Yes
DF=Deterministic forecast        N=No

## 3.1 Germany

*General FFEWS setup*

At present, forecasting and warning of hydro-meteorological events in Germany is split between the German Weather Service (DWD), flood forecasting centers in the Federal States, and flood emergency management in the counties (Landkreis). The DWD issues weather forecasts and provides severe weather warnings. Flood forecasting centers translate the upcoming weather into hydrological forecasts and provide their interpretation into warning messages. In April 2024, the Federal Act on the DWD (Gesetz über den Deutschen Wetterdienst) was changed so that the DWD is now allowed to also disseminate warnings other than severe weather warnings, which include flood warnings. A national natural hazards online portal is currently under development; its release is expected in 2025. Local emergency management authorities interpret the information from forecasting agencies and translate it to impact-specific warnings and are responsible for decision making and triggering specific emergency actions, such as evacuation. Warnings are disseminated to the public via the centralized Modular Warning System (MoWaS), operated by The German Federal Office of Civil Protection and Disaster Assistance (BBK).

*Meteorological forecasting*

In Germany, the DWD is mandated by law to provide the official weather forecasts and severe weather warnings. Their ICON-D2-EPS model provides a 20-member ensemble forecast for the upcoming 48 hours. The DWD has developed a so-called "seamless" forecasting system to better forecast small-scale convective events and flash floods, called SINFONY. This system integrates ensemble weather forecasts and nowcasting techniques (based on real-time radar measurements) over short lead times (Blahak et al., 2024). Forecasts from DWD are delivered to the environmental offices and flood forecasting agencies of the Federal States and to water associations ('Wasserverbände'). These contain a description of the characteristics of severe





weather events and the amount of expected precipitation. The thresholds and warning levels for heavy rainfall as used by the
DWD are summarized in Fig. 3.

*Hydrological forecasting*

Ensemble weather predictions are used as input for hydrological flood forecasting models at 10 flood forecasting centers in
the Federal States. Additionally, regional flood forecasting centers, specifically for large navigable rivers such as the Rhine,
are in operation. Data exchange between flood forecasting centers and neighboring countries is implemented, for example by
sharing gauging station data. An overview of flood forecasting models, and the locations of gauging stations at which the
hydrological discharge and water level forecasts are issued are provided in the Report of the German LAWA-Commission
(LAWA, 2020). The national web portal (Table 1) was established after the severe flood in August 2002 and provides an
overview of gauges with up-to-date flood forecasting and warning information. Moreover, it serves as a gateway to the specific
web services of flood forecasting centers in the Federal States. It also disseminates written interpretations of the forecasts (in
so-called "Lageberichte"). The LAWA (2020) report further contains a detailed overview of the frequency of forecast updates
for specific gauges in case of flooding, which typically ranges between 2 and 24 hours.

Here, we focus on the regional flood prediction in the German Federal States of NRW and RLP (Fig. 2). The environmental
office in RLP (Landesamt fur Umwelt, LfU) is responsible for flood forecasting. They issue forecasts and warnings with a 48-
hour lead time, based on a 20-member ensemble forecast from the DWD ICON-D2-EPS model. The forecasts are updated
every three hours (same frequency as the DWD forecasts). In NRW, the responsibility for flood forecasting belongs to the
Landesamt für Natur, Umwelt und Verbraucherschutz Nordrhein-Westfalen (LANUV). Currently, water level observations at
some gauges in NRW are accessible in near-real-time, but forecasts are not publicly available (Table 1). In NRW, LANUV
runs hydrological ensemble forecasts operationally every 3 hours for 10 days in the future. These predictions are based on the
Large Area Runoff Simulation Model (LARSIM) hydrological model and use weather forecast data from the ICON-D2-EPS
(20 ensemble members) and ICON-EPS (40 ensemble members) models. The two regions, NRW and RLP, use different color
codes and warning levels (Fig. 3). While the warning thresholds in RLP are based on water level return periods, the thresholds
in NRW are based on local impacts to buildings and communities. NRW defines four different warning levels, while RLP uses
six. NRW is in the process of adding an additional purple warning level as well, which represents a risk of catastrophic floods.
This would better streamline the hydrological warnings of NRW and RLP and the meteorological warnings from DWD, which
also include a purple level (Fig. 3). Besides some prototypes in pilot areas, near-real-time flood inundation and impact
forecasting are not yet implemented in the operational flood forecasting and warning systems in Germany (Merz et al., 2020).

*Crisis management and communication*

Warnings from the federal hydrological flood forecasting centers (e.g. LANUV in NRW) are sent to the county (Landkreis)
crisis response centers. Based on laws regarding general civil assistance and disaster management all counties, municipalities,





and free cities in Germany, which are not belonging to counties, are required to have a flood warning and preparedness plan (DKKV, 2024; Ministerium des Innern und für Sport Rheinland-Pfalz, 2020). For example, RLP developed the Rahmen-,

Alarm- und Einsatzplan Hochwasser (Ministerium des Innern und für Sport Rheinland-Pfalz, 2020). The county crisis response centers (not the federal state hydrological forecasting centers) are responsible for dissemination of the warnings to the public. In practice, while these plans are required it is only in the aftermath of a disaster that it is revealed whether the plan is realistic, can be implemented, and is effective for crisis management in the county (DKKV, 2024).

Both the DWD and flood forecasting centers can activate MoWaS in case of an expected catastrophic flood and further disseminate information to other channels such as warning apps (Meine Pegel, NINA or KATWARN), radio, and television (IIASA, 2022). Since February 2023, the population can be warned using a new cell broadcasting system, which does not require the installation of an application and is connected to the MoWas system. In terms of online information, residents show a clear preference for maps in warning messages (Lindenlaub et al., 2024a).

**3.2 Luxembourg**

*General FFEWS setup*

Meteorological forecasts are used by the flood forecasting department (Service de prévision des crues) of the water management authority (Administration de la gestion de l'eau, AGE). The AGE produces forecasts for the main rivers systems (the Alzette, Chiers, Moselle, Sûre and Syre), but also for the Mosel in France, which are publicly available via the web portal

(Table 1). In case of a large flood, a crisis team is activated to coordinate emergency management activities. Civil protection activities and the organization of the response during normal floods are conducted by the national fire brigade: the CGDIS (Corps Grand-Ducal D'Incendie et de Secours). Recent developments include the LU-Alert warning system for mobile phones. In addition, warning levels and color codes have been recently streamlined between the meteorological and hydrological forecasts, and the crisis management.


*Meteorological forecasting*

The national meteorological service Meteolux does not run a separate weather model but instead uses different weather models from neighboring countries (ECMWF, DWD, MeteoFrance). A severe weather risk assessment unit (Cellule d'évaluation du risque intempéries, CERI) can be initiated to estimate the societal impacts. The CERI includes representatives from the AGE,

MeteoLux, the national government (Haut-Commissariat à la protection nationale), and CGDIS. Meteolux recently adopted new warning levels: yellow-orange-red-purple (Fig. 3). The largest change is the inclusion of a purple warning. This can generally only be issued in case of severe flash floods with imminent danger, requiring immediate action.




*Hydrological forecasting*

Hydrological forecasts and warnings are provided by the flood forecasting department of AGE. They use a wide range of meteorological forecasts as input, from the DWD, ECMWF and MeteoFrance. The forecasts are made with the LARSIM model. They have a lead time of 24 hours for smaller streams, and 48 hours for the large rivers (e.g., Moselle, lower-Sauer).

They are generally updated every three hours during normal situations and every hour during emergencies. AGE only publishes the deterministic forecasts on the portal, while multi-model ensemble forecasts are used internally. Warning levels in Luxembourg have recently been revised, and AGE now uses three different levels: yellow ("Cote de vigilance"), orange ("Cote de pré-alerte") and red ("Cote d'alerte"). Those are in line with the warning levels of Meteolux, and the crisis management (see below). However, the hydrological service (AGE) is not mandated to issue a purple warning in the new system. This is

reserved for the meteorological forecasting service or crisis management. Interviewees from Luxembourg stressed that the new warning levels are based on expected impact, derived from flood maps and/or consultations with local authorities and firefighters (Fig. 3).

*Crisis management and communication*

During regular floods, flood response is managed by CGDIS, who are in constant exchange with the flood forecasting service. When the flood risk is deemed high, a Crisis Cell is activated, led by the High Commission for National Protection (HCPN) and comprising representatives from various bodies, such as the Police, Army, and Government. During a crisis, the AGE flood forecasting department is in constant exchange with CGDIS and the High Commission for National Protection (HCPN). The HCPN manages the actions required during a crisis (European Commission, 2024). Four crisis management phases are

specified: normal, vigilance, pre-alert, and alert (Le gouvernement du Grand-Duché de Luxembourg, 2024). Those are in line with the hydro-meteorological warnings described above. In the 'pre-alert' phase, local flooding is expected and in the 'alert' phase widespread flooding is expected with a significant impact on people and property (Le gouvernement du Grand-Duché de Luxembourg, 2024). These alert phases are also coupled to specific water levels. In the '(pre-)alert phase', flood bulletins are circulated which include warnings and a textual interpretation. At the 'alert phase', the prime minister (or his/her delegate)

can activate a Crisis Unit Cell (Le gouvernement du Grand-Duché de Luxembourg, 2024). The CGDIS pointed out that the procedures have greatly improved since the July 2021 floods. For example, before the 2021 flood event, communication between municipalities and CGDIS often failed. Nowadays, flood alerts are sent to the local fire stations along with a request to contact the (deputy) mayor of the municipality.

To improve warnings to the public, the new LU-alert system has recently been launched (on 17 October 2024), which supports both cell broadcasting and geo-located SMS services (Le gouvernement du Grand-Duché de Luxembourg, 2024). The SMS service has the additional advantage that it enables to monitor the number of people in a certain area, which CGDIS can use to monitor how many people have been evacuated.





### 3.3 The Netherlands

*General FFEWS setup*

The Royal Netherlands Meteorological Institute (KNMI) is responsible for the severe weather warnings in The Netherlands. KNMI and national hydrological forecast information is transmitted to the regional authorities, the Water Boards, who are responsible for forecasts in the smaller streams and catchments. They advise the crisis management organizations (the so-called 'Safety Regions') about the risk and the implementation of protection measures.


*Meteorological forecasts*

Among the meteorological models used are the HARMONIE-AROME (Frogner et al., 2019) and ECMWF (Haiden et al., 2018) weather models. The HARMONIE model provides hourly, and the ECMWF model 6-hourly forecasts. The ensemble model-version of HARMONIE is called HarmonEPS, which produces a subset of ensembles at every hourly run ('a running 320 ensemble'). This is different from ECMWF EPS, which runs full ensembles every 6 hours. Meteorological warnings are classified into the yellow-orange-red color code (Fig. 3). The KNMI developed an in-house system to calculate maps of exceedance probabilities for an ensemble weather forecast: the Probabilistic Alert System for the Concurrence of Adverse Weather Elements (PASCAL) system (Kok, 2022).

The KNMI works towards impact-oriented forecasting. To enhance the knowledge and experience on those impact-oriented forecasts, the KNMI launched an Early Warning Centre in 2015. In case an orange or red warning is predicted by the ensemble models (> 60% probability), the KNMI initiates an online multi-disciplinary weather-impact team meeting. This team includes key organizations such as ProRail, the National Crisis Centre (NCC), the Netherlands Traffic Center (VCNL) and the National Operational Coordination Centre (LOCC). This team provides advice to the KNMI on the selected warning level, which is in 330 practice always followed. The stakeholders in the team also fill in an impact questionnaire, which results in an impact score per economic sector. The final warning level is decided by the operational lead at the KNMI. To allow for an early engagement of the impact team, the KNMI can already inform the impact team up to seven days in advance when a warning is not yet generated, but possible (using a 30% probability threshold). In case a rainstorm is predicted only at the last moment, the KNMI issues a warning without first consulting the impact team.


*Hydrological forecasts*

The national Dutch Water Management Center (WMCN) produces 5-day (validated) and 15-day (automatic) forecasts for the river Rhine (gauge Lobith) and river Meuse (gauge Sint-Pieter). Forecasts 5-days ahead are deterministic, while the 15-day forecasts are ensemble products. These forecasts use multiple weather models as input: DWD COSMO-LEPS model, DWD 340 ICON-EU (5-days), DWD ICON (5-days), the ECMWF Ensemble Prediction System (EPS), and ECMWF HRES (deterministic) forecasts. Meteorological uncertainty is included by using ensemble forecasts, where every ensemble member



is used as input to the hydrological model. For a small portion of stations (approximately 5%), a post-processing technique is used on the discharge forecasts (Verkade et al., 2017) to include the hydrological uncertainty. This is done by correcting ('dressing') an individual streamflow ensemble member based on historical hydrological model errors. In addition, the WMCN
uses European EFAS forecasts and additional information from neighboring countries.

The Dutch national protocol for high waters and flooding (LDHO) provides an overview of the different warning levels for the Dutch main rivers (WMCN, 2023). Four warning levels are in place (Fig. 3): green, yellow (elevated water levels, water nuisance possible), orange (high waters) and red (extreme high water). These are based on both water level and discharge
thresholds, and expressed as return periods (WMCN, 2023, Appendix D). Different return periods apply to different rivers, as the strength of the dikes is different (WMCN, 2023). Therefore, Fig. 3 displays ranges of return periods. The exact return periods per river can be found in WMCN (2023, Appendix D). As multiple ensemble forecasts are made (using multiple models), expert judgement is needed to determine which warning level is eventually selected.

In case of orange or red warnings, an expert group gathers into a national committee: the Landelijke Commissie Overstromingsdreiging (LCO). These experts in meteorology and hydrology translate the water level forecasts into impacts and advise on required interventions. The LCO is mandated to evaluate the color code selected by WMCN and can increase the warning level if deemed necessary (e.g., in case of calamities, such as failures of water infrastructures). As a result of these decisions, information in a web-portal (Table 1) is updated and disseminated every day, which includes the hydrological
forecasts, warning levels, and a textual interpretation. In case of orange and red warning levels, the online information can be updated multiple times per day. The orange warning level can be issued with a maximum lead time of 48 hours. However, the stakeholders can be informed up to 7 days in advance with a notice that the threshold will potentially be exceeded.

At the local scale, the Water Boards and Safety Regions are responsible for hydrological forecasting and responses,
respectively. For example, the Water Board of Limburg runs operational forecasts to predict discharge and/or water levels, with a lead time of 5 days. Predictions are deterministic, although a strong need and wish exists to deploy probabilistic forecasting approaches. After the 2021 floods, the Water Board of Limburg developed clear warning thresholds for the smaller rivers (Veiligheidsregio Limburg-Noord & Zuid-Limburg, 2023). Those are identified based on the expected nuisance to the (urban) surroundings and thus take the strength of the dikes and the exposed assets into account (Veiligheidsregio Limburg-
Noord & Zuid-Limburg, 2023).

*Crisis management and communication*

The Safety Regions are regional organizations responsible for crisis management for various kinds of hazards and the provision of information to the population. The Netherlands is the only country with such administrative units specifically designed for
crisis management. Based on the estimation of LCO and the Departementaal Crisis Centrum (DCC) of the Ministry of



Infrastructure and Water Management, the Safety Region and other stakeholders can be warned through the National Crisis Management System (LCMS). Actions are linked to the warning levels (e.g., dike inspections), which are detailed in WMCN (2023) for the large rivers and in the protocols from the Water Boards for the small rivers. Some regions in The Netherlands have a dedicated emergency management plans for floods. Limburg, the Dutch province, which was most heavily affected by 380 the 2021 floods, developed an extensive flood emergency management plan (Rampbestrijdingsplan Hoogwater Limburg, Veiligheidsregio Limburg-Noord and Zuid-Limburg 2023). After the 2021 floods, also the small rivers (e.g., Geul) were included in this plan. In case of a flood crisis, the departmental (DCC) or national crisis management center (NCC) coordinates the crisis, and the LCO advises on measures to take (WMCN, 2023).

The public is warned through multiple channels, of which the most important are NL-ALERT (Table 1) and the 4278 air raid alarm systems. The KNMI also actively disseminates warnings through the newly developed KNMI app, and through social media. They have a clear impact- and action-oriented warning approach (see Section 4.5).

### 3.4 Belgium

*General FFEWS setup*

Meteorological forecast and warnings for Belgium are produced by the Royal Meteorological Institute of Belgium (RMI). River discharge forecasts are separately generated for Flanders and Wallonia, using RMI forecasts as input. The National Crisis Center (NCC) is responsible for managing national level crisis situations.

*Meteorological forecasts*

The meteorological forecasts are produced by the ALARO (Gerard et al., 2009) and HARMONIE-AROME (Frogner et al., 2019) models. RMI is currently developing an ensemble prediction system RMI-EPS, which consists of a combination of AROME and ALARO with 11 ensemble members each (Smet, 2017). However, operational meteorologists also use models from other weather centers such as the UK Met Office, DWD, and ECMWF.

Warnings are generated using the yellow-orange-red color codes on a provincial level (Fig. 3). Warnings are issued if the 65% probability threshold is exceeded for >25% of the area, or the 65% probability threshold is exceeded of the following color code for < 25% of the area, or the 15-65% probability threshold is exceeded of the following color code for > 25% of the area. These are guidelines, but the RMI can deviate from those based on the expected impact. The impacts are discussed in online impact team meetings, which can be called 2-3 times a day during severe weather conditions. These meetings involve the 405 hydrological centers for Wallonia and Flanders (SPW and HIC, respectively), fire brigades, regional and national crisis centers and provincial governors. Although the RMI takes impacts into account in selecting the warning levels (e.g., a rainstorm approaching a festival), the interviewee stressed that the RMI is not mandated to provide impact- or action-oriented advisories. Four radars provide near real-time observations for nowcasting using the INCA-BE system. This system allows for the



generation of so-called "flash warnings", which are automatically generated warnings at the local level (i.e., municipal) with
<1h lead time (Smet, 2017). These warnings are disseminated via the RMI app.

*Hydrological forecasts*

In Flanders, the Hydrologische Informatie Centrum (HIC) is responsible for the monitoring and forecasting of navigable
waters. The non-navigable rivers (category 1) in Flanders are monitored and forecast are issued by the Vlaamse
MilieuMaatschappij (VMM). Those observations and forecasts are publicly available on the official Waterinfo web-portal
(Table 1). The HIC produces long-term (10 days ahead) and short-term forecasts (48 hours ahead) using the hydrological
model FEWS-Flanders. The long-term forecasts are based on deterministic ECWMF forecasts (2 times daily), while the short-
term forecasts are derived from deterministic HARMONIE model runs (4 times daily). The HIC has four hydrodynamic models
for different catchments: (1) Leie-Bovenschelde Gentse Kanalen, (2) Zeeschelde, (3) Zenne-Zeekanaal, (4) Demer. These
models provide inundation forecasts, which are also visualized in the web-portal and disseminated to partners. Apart from the
river Demer, HIC runs the FEWS system for Flanders 4 times per day with input from the ALARO-model of RMI. The rainfall
forecasts used as input for the hydrological model are deterministic. Probabilistic discharge forecasts are made using a
hydrological post-processing technique, based on historical lead-time dependent model errors (Van Steenbergen et al., 2012).
Two types of warnings can be generated 48 hours in advance for both the navigable and non-navigable rivers: alarm states and
flood states. These warnings follow the green-yellow-orange-red color codes. The flood warning states (Fig. 3) are visualized
on so-called 'Vrijboordkaarten' and show the distance between the water level and the dike crest. The colors reflect the
following situations: green: no floods; yellow: pre-warning; orange: warning, non-critical flooding would be possible; red:
alarm, critical flooding would be possible. These flood warning states are based on water-level thresholds determined by the
hydrological-hydraulic model forecasting results. The alarm states on the other hand are determined based on the impact on
water structures (e.g., bridges), but do not reflect overall flood impacts to society. The forecasts and warning levels are
displayed in near real-time on the Waterinfo portal (Table 1). The warnings are disseminated to the De Vlaamse Waterweg
waterways authority, which distributes the warnings to other stakeholders and emergency management.

In Wallonia, the Direction de la Gestion Hydrologique (DGH) of the Walloon government public service (Service Public de
Wallonie, SPW) is responsible for the monitoring, forecasting and warnings. The DGH operates the Hydromax model using a
network of 150 river gauging stations and 100 rain gauges (the 'Wacondah' network) as input (DGH, 2024). The warnings
follow a green-yellow-red color code (Fig. 3) and are issued on a catchment level. Yellow warnings are issued if one (or more)
rivers in a basin are expected to cause localized and non-severe floods, and red warnings if one or more rivers of a basin are
expected to cause major floods with an impact on infrastructure and local residents. The thresholds are not prescribed but
defined by expert judgement depending on catchment wetness state, precipitation forecasts, discharge in upstream rivers and
tributaries, and from past experience. The warnings do not take the small tributaries into account. These smaller tributaries
(e.g., the Vesdre) are monitored by The Direction des Cours d'Eau Non Navigables (DCENN), which operates a measurement





network called "Aqualim" (DGH, 2024). None of the governmental hydrological and crisis management agencies in Wallonia produce and/or use inundation forecasts operationally. Nonetheless, inundation forecasts are produced for some pilot river
sections; but currently these forecasts are not communicated to the public since they do not consistently cover the whole river network.

*Crisis management and communication*

Crisis management in Belgium is organized at three levels: municipal, provincial, and federal state level. At the federal level
crisis management is coordinated by the Centre de Coordination des Risques et de la Transmission d'Expertise (CORTEX) for Wallonia and the Crisiscentrum van de Vlaamse overheid (CCVO) for Flanders.

In Wallonia, in case of an orange warning, CORTEX calls in a special flood expertise unit (CELEX) by videoconference. In case of (pre-) alerts, CELEX develops and shares flood reports with stakeholders (but not with the general public), which
include an outlook of the consequences and required actions. This process started after the 2021 flood. In case of a large flood disaster, such as in 2021, the federal level is initiated and the National Crisis Center (NCCN) takes over the coordination. The Royal Meteorological Institute (RMI) can directly send meteorological warnings to the NCCN. Furthermore, every municipality in Wallonia needs to have a general emergency management plan, which can also be developed for floods (Plan Général d'Urgence et d'Intervention; PGUI). However, these flood-specific plans are not mandatory. Since 2017, the public
can be warned via mobile cell broadcasting system: BE-ALERT (Table 1).

## 4. Key challenges of operational FFEWS

The interviews revealed that the FFEWSs in all regions are under strong and rapid development after the 2021 floods. All regions now use probabilistic forecasts for both rainfall and streamflow, have emergency response plans for floods, and use cell-broadcast services for alerting the population. Moreover, all regions defined clear thresholds for the different hydrological
warning levels, except for Wallonia. As a response to the 2021 floods, a new emergency response plan for Limburg ("Rampbestrijdingsplan Hoogwater Limburg") now includes flood scenarios for small rivers (e.g. Geul river), including clear warning thresholds and corresponding actions. Luxembourg recently implemented a new alerting system, harmonizing the alerts from the meteorological and hydrological forecasts, as well as the crisis management. While many of such improvements take place, the interviewees identified several issues required for further improvement of FFEWSs.

## 4.1 Streamlined thresholds /warning levels

In general, the warning levels and color codes of meteorological and hydrological levels not only differ between countries but even within countries (Fig. 3). The amount of warning levels and their color codes are different in the two federal states of Belgium: Flanders and Wallonia (Fig. 3). Also in Germany the warning levels are not consistent among Federal States. For



example, in RLP they are linked to flood return periods and in NRW to flood impacts (NRW, 2024) (Fig. 3) – the latter not

necessarily corresponding to the same return periods as in the neighboring RLP (DKKV, 2024). While RLP now distinguishes six warning levels, other German states use just four (like NRW), including a green level for no warning (Fig. 3). Interviewees in RLP also state that the number of alerts by the German meteorological office (DWD) have been high in the past during some periods of possible events and that it could be difficult for the recipients to estimate if a warning indeed results in a flood. Furthermore, DWD warnings are generally cleared at the end of a precipitation event, while hydrological warnings can still be

active because the peak water levels in the rivers are not yet reached. According to the interviewees, such a situation can be confusing for local stakeholders who are not aware of the time lag between rainfall and runoff processes in the catchments. Research shows that inconsistent visual and textual warnings from multiple sources are confusing the public and lead to ineffective communication and actions (Weyrich et al., 2019).

## 4.2 Forecasts for catastrophic events

After the 2021 flood, some regions started a discussion to introduce a new (or additional) color code for extreme floods. This discussion has led to selecting 'dark purple' as the color for the highest warning level in the German Federal State of RLP, based on the 100-year return period threshold. The warning levels for Luxembourg also contain a new purple level, which can only be issued by the meteorological service MeteoLux and not by the hydrological service AGE (Fig. 3). The highest warning level of AGE is "Cote de vigilance rouge" (Red), which is often linked to small return periods. For example, gauge Bollendorf

in the Sauer has a red warning threshold related to a water level of 4.25 m, which is less than a 5-year return period event. This suggests that extreme riverine floods could be better represented in the new warnings. The Belgium region Wallonia has also started a discussion on an additional warning level and states that "...there should be a new discussion about the need for an extra level of warning (e.g., dark red) for really extreme situations". For the Limburg Water Board, the highest warning level (red) is described as a "water nuisance" to people, rather than a high-impact catastrophic event. However, more extreme

thresholds come with more uncertainty. Rare events will typically be forecast with only a few ensemble members. These low-probability and high-impact forecasts are difficult to cope with for decision-makers (Speight et al., 2021), as was also observed for the 2021 event. There is a need to better understand such extreme events, to better represent them in the models and forecasts, and to better predict potential impacts. This was raised by respondents in all countries.

Flood hazard maps can be used to estimate which areas are going to be flooded during an expected event (e.g., Dottori et al., 2017). However, the use of static flood hazard maps showed large deficiencies during the 2021 flood. Vorogushyn et al. (2022) showed that the inundation areas in 2021 exceeded the most extreme official hazard maps in the Ahr valley and recommend developing more severe extreme scenarios. In Wallonia, responders and households in 'safe green or unmarked zones' faced the 2021 flood and did not know what to do. Many of the flooded areas directly along the Vesdre river (a tributary of the

Meuse river) were marked as safe green zones or simply left unmarked in official flood hazard maps. As a result, many





residents were not aware of the flood risk, and were surprised their houses were flooded up to the second floors (Rodríguez Castro et al., 2025).

## 4.3 Impact-based forecasting

Impact-based warnings provide crisis managers and first responders with more tailored information (Apel et al., 2022; Boult
et al., 2022; Red Cross Red Crescent Climate Centre, 2020; Meléndez-Landaverde et al., 2020, Merz et al., 2020, Rhein and Kreibich, 2025, Poolman et al., 2018). However, the practice of impact-based forecasting is still in its infancy (Merz et al., 2020). Only about 30% of the national meteorological and hydrological services in Europe use impact-based models (Kaltenberger et al., 2020; Schroeter et al., 2021).

Our study confirms that the operational uptake of impact-based forecasting is challenging, but crucial. For example, the hydrological service in Wallonia emphasizes the need for impact-based forecasting and states that currently the same warning level for different events can mean different impacts on the ground (DGH, 2024). Many interviewees stressed that estimates of potentially flooded areas are highly needed to improve the effectiveness of early actions. Citizens showed a clear preference for inundation maps in warning messages they receive, compared to other information types (e.g. a warning symbol;
Lindenlaub et al., 2024a). Moreover, people tend to assess the flood magnitude higher when an inundation map is provided (Lindenlaub et al., 2024b). The high value of detailed flood inundation predictions for emergency management is also emphasized in England (Aldridge et al., 2020). However, it is not yet clear whether warning messages accompanied by information on flooded area lead to better protective behaviour of residents (Kuller et al., 2021). The HIC in Flanders is the only operational center included in this study which produces inundation forecasts operationally (visualized in their portal,
Table 1). However, these inundation forecasts are not used in any of the official warning levels (see Fig. 3). Recent computational advances in flood modeling, such as the use of machine-learning, Graphical Processing Units (GPUs) and cloud computing make real-time inundation modelling realistic and affordable (Hofmann et al., 2021; Speight et al., 2021; Apel et al., 2022; Agerbeek et al., 2024). Apel et al. (2022) produced inundation forecasts for the 2021 flood event in the Ahr valley at 10 m spatial resolution in a timespan suitable for operational application (14 min on a GPU) and showed the strong added
value of these forecasts for operational crisis management. Estimates of inundated areas (and other factors, e.g., flow velocity) can be overlaid with exposed buildings and critical infrastructure, to support rapid mapping of flood impacts (e.g., Apel et al., 2022, Najafi et al., 2024 and Dottori et al., 2017).

Despite the challenge of operational impact-based forecasting, some best practices in the region should be outlined. Some
regions, such as NRW in Germany, developed impact-based warning thresholds (Fig. 3). Here, the water level thresholds are chosen based on the expected impact to assets and people. To translate hydro-meteorological forecasts to impacts and advisories, countries established flood expert teams which interpret and translate the forecasts to impacts and advisories for the local authorities. CELEX in Wallonia and the LCO in The Netherlands are examples of such groups, which provide an





interpretation and translation of the meteorological and hydrological forecasts. The KNMI is spearheading the development of
impact-based warnings through the development of the Early Warning Centre, involving a range of multi-sectoral stakeholders.
The KNMI also emphasized that their efforts on impact-based forecasting contribute to stronger partnerships and
collaborations, with organizations which are all working towards the same goal of predicting (flood) impacts (Red Cross Red
Crescent Climate Centre, 2020).

## 4.4 Coordination and data sharing across borders

All interviewees emphasized that data exchange between countries is already quite well organized. For example, the border
between The Netherlands and Flanders is marked by the lower part of the Meuse river ('Grensmaas'). To provide a consistent
warning for all households along the Grensmaas, Flanders adopts the warning levels provided by the Dutch Government. In
addition, the Dutch water management center (WMCN) uses both the European data from the EFAS system and additional
hydrological information from neighboring countries. Germany also installed an automatic data exchange between flood
forecasting centers from its neighboring countries. Some data exchange across countries faces challenges due to differences in
the levels of responsibility for disaster coordination and communication. For example, better cooperation on modelling and
the development of common models is needed to ensure consistency and provide tailored information within the same river
system at different times, based on stakeholders' requirements. Furthermore, HIC in Flanders specifically suggests improving
the exchange of information on upstream reservoirs in Wallonia. Interviewees from Waterschap Limburg stressed the
importance of an international exchange of flood warnings, which is currently lacking.

## 4.5 Communication of actionable information

One of the interviewees mentioned: *"Forecasts are not the problem anymore, the problem is to act"*. Dasgupta et al. (2024)
suggest that a key factor for inaction is ineffective communication of the forecasts, including their uncertainty. Here we
elaborate on this challenge, specifically for communication to crisis managers and the local public.


### *Communication with crisis management*

Interviewees stressed that the information needs for emergency management are regularly not met. Interviewees in Wallonia
noted that not all municipalities have specific flood emergency management plans, even if they are in areas of high flood risk.
In The Netherlands, local Waterboards (responsible for regional and local water management) and Safety Regions
(organization of first responders) stressed the challenge of interpreting the issued forecasts by national services (KNMI and
WMCN). An evaluation of the 2021 floods in Germany already pointed out that there is a lack of actionable information in
forecast, which is highly needed for first responders and the population (DKKV, 2024; Thieken et al., 2023a). Our interviewees
in Germany pointed out that, although improvements are seen since 2021, the information needs for triggering actions for
responders are still not met. They require more specific and detailed information, such as flood inundation and damage
predictions. Also, in Wallonia during the 2021 flood, the crisis managers requested information about the expected flood



inundation extents which could not be provided. This has eroded trust between different stakeholders. A loss of trust is detrimental for the functioning of the early warning system, as it hampers the coordination and the response in the field (Seebauer and Babsicky, 2017).

Multiple interviewees point to deficiencies in the communication chain of forecast agencies to crisis managers. In Flanders, the warnings from HIC are sent to the De Vlaamse Waterweg waterways authority, who distribute it further to other stakeholders. However, flood risk management is not a core business of the De Vlaamse Waterweg authority, and therefore, they lack specific expertise on flood risk. Consequently, warnings are sometimes misinterpreted and not well communicated further. Similar findings are reported from experts in Luxembourg, where communication between the crisis management
(CGDIS) and hydro-meteorological agencies (MeteoLux and AGE) is sometimes challenging. Interviewees from Luxembourg suggested that a central Early Warning Center for Luxembourg will be highly valuable to condense the different forecasts into actionable warnings. In The Netherlands, evaluations after the 2021 flood show the need for centralized and streamlined information (COT, 2022). Our interviewees in The Netherlands stress that local stakeholders (e.g., municipalities) still suffer from fragmented pieces of information from multiple stakeholders such as the KNMI, water boards and the WMCN.


Research shows that probabilistic information can lead to better decisions (Verkade & Werner, 2011). Most operational forecasting centers in the region use probabilistic forecasting (Table 1). However, probabilistic approaches are often lacking for smaller rivers in The Netherlands. Moreover, the communication of probabilistic forecasts can be challenging (Arnal et al., 2019). In Germany, Luxembourg and the Netherlands it was mentioned that local emergency responders often prefer one value
(e.g., water level) to take action upon. However, this value can be misleading due to the inherent uncertainty. Therefore, it is crucial to invest in better communication and interpretation of probabilistic forecasts and translate them into action advice. The Netherlands is advancing on the communication of probabilistic forecasts by providing tables with exceedance probabilities for each discharge threshold (WMCN, 2024). It is recommended for other forecasting centers to follow this approach.


### *Communication to the local public*

Germany and Luxembourg installed a cell broadcasting system in response to the 2021 flood. Now all countries covered in this study have such systems in operation (Table 1). Despite the effectiveness of cell broadcasting, research shows the need of a multi-channel approach due to the risk of failure of one system (e.g., power shortages or hacks) (Mahdavian et al., 2020).
Air raid sirens are another effective means of mass-alerting the population. The countries in our study have a different perspective with regards to the air raid sirens. The national fire brigade of Luxembourg (CGDIS) emphasized the importance of a multi-channel alerting approach, including the air raid sirens. However, in Wallonia the sirens are not used, and The Netherlands plans to stop the system, mainly because of the high maintenance costs which due outweigh the societal benefits



(Ministerie van Justitie en Veiligheid, 2024). In Germany, the sirens are being reestablished. The different approaches of
neighbouring countries to mass-alerting the population in case of a flood or other catastrophe are remarkable.

People take more (effective) protective actions if warnings are impact-based (Meléndez-Landaverde et al., 2019), and even
more if those warnings also include behaviour recommendations (Red Cross Red Crescent Climate Centre, 2020; Golding et
al., 2019). The German national flood portal (https://hochwasserzentralen.de, Table 1) recently included action
recommendations as addition to the hydrological warnings. The Dutch meteorological office KNMI recently hired a
communication expert (specialized in social media) to produce clear infographics on '*what can you expect*' and '*what can you
do*' during emergency situations (KNMI, 2024). This is presented in plain language (B1 level). We identify this as a best
practice that can be followed by other regions. Multiple organisations during our interview stressed that they lack capacity or
knowledge to develop effective communication protocols to inform the public. Multiple interviewees (e.g., KNMI and AGE)
stressed the importance of tailoring the communication strategy to different target groups. For example, young people in The
Netherlands receive information through social media, such as Instagram, and generally will not download the KNMI phone
application.





## 5. Conclusions and recommendations

This study reviews the status of the Flood Forecasting and Early Warning Systems (FFEWSs) in the countries in western Europe hit by the July 2021 flood: Germany, Luxembourg, Belgium, and The Netherlands. Expert interviews over the region reveal that all systems are under strong and rapid development after the 2021 flood event. As a result, all countries now have probabilistic FFEWSs and, except for Wallonia, all regions have clear pre-defined warning thresholds based on peak discharge or water levels. In most regions, hydrological warning thresholds are determined based on expected local flood impacts.

Moreover, over the past years, online flood forecasting portals have been improved, emergency response plans updated (with a stronger focus on small rivers), and national scale cell broadcasting is implemented to send out phone-based alerts in case of immediate danger.

Strong differences exist in flood warning levels and color codes across and within the countries. Additional research is

necessary to determine whether an international harmonization of the warning levels is desirable. As a response to the extreme flood event of 2021, Luxembourg and some regions of Germany have recently introduced an additional purple warning code for the most extreme weather and hydrological events. In countries where red is the maximum warning level, some experts suggested also adding a dark purple level to represent truly catastrophic impacts such as the 2021 floods. Despite having an extreme warning level can be helpful for indicating the potential disastrous consequences of an event, it is still under debate

whether more warning levels supports a more effective communication to the public and responders.

The implementation of operational impact-based forecasting and warning is challenging. Although impacts are sometimes considered in the design of the hydrological warning thresholds, the thresholds are often too conservative (too low) and thus do not reflect major societal flood impacts. Moreover, meteorological, and hydrological forecasts often lack specific impact

and action information, which is highly needed for first responders at the local level. The forecasts of river discharge, return periods, and corresponding warnings are often hard to interpret by local decision-makers, emergency services and local population. To trigger effective early actions, it is strongly suggested to enrich forecasts with impact-based information and to provide reliable predictions of inundated areas and risk hotspots. Flanders is the only region where operational flood inundation (i.e., flood maps) forecasts are provided. Recent computational developments in inundation modelling such as cloud

computing, GPU-based parallelization and machine-learning will significantly reduce run times and pave the way for the operational implementation of inundation forecasts.

Our study recommends expanding resources on impact-based forecasting and effective communication protocols, including tailored action advisories. This in turn will require stronger collaboration between the forecasting agencies and emergency

management authorities. More knowledge and insights are urgently needed to better understand the needs of forecasting agencies to accelerate impact-based forecasting. Moreover, it needs to be fully understood what kind of impact and action



information emergency managers and civilians need to make more effective decisions. These challenges need to be addressed to reduce the gap between early warning and early action for impactful flood events.

## Author contribution

**TB**: Conceptualization, Investigation, Methodology, Writing, Visualization, **JA**: Conceptualization, Investigation, Methodology, Writing, Funding acquisition, Project administration, **DRC, BD, SV, RL, JK, DZ, HM, AT**: Conceptualization, Investigation, Methodology, Writing, **KS:** Conceptualization, Investigation, Methodology, Funding acquisition, Project administration, **PW**: Investigation, Writing, **LF**: Writing, **JV:** Conceptualization, Investigation, Methodology.

## Competing interests

The authors declare that they have no conflict of interest.

## Acknowledgements

This research has been supported by the JCAR ATRACE project (https://www.jcar-atrace.eu/). We gratefully thank KNMI, (The Netherlands), SPW (Belgium), LfU (Germany), RPTU (Germany), WMCN (The Netherlands), HIC (Flanders, Belgium), AGE (Luxembourg), RMI (Belgium), SPW (Wallonia, Belgium), Waterschap Limburg (The Netherlands), LANUV
(Germany) and CGDIS (Luxemburg).

## Financial support

This research was supported by the JCAR-ATRACE project. The JCAR-ATRACE project has received funding from the Dutch Ministry of Infrastructure and Water Management with project no. 012215.



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



## Appendix A. Interviewed specialists

### Netherlands

- Senior operational flood forecasting expert at **WMCN**
- Senior operational early warning specialist and program manager at **KNMI**
885
- Senior flood early warning specialist at **Waterschap Limburg**

### Belgium

- Professor at University of Gent and lead of the meteorological forecasting department of **RMI**
- Senior operational flood forecasting expert at **HIC Flanders**
- Two senior crisis managers at **SPW**

890 ### Germany:

- Two senior operational flood forecasters at **LfU**
- Senior operational flood forecaster at **LANUV**
- Professor at University of Kaiserslautern-Landau (**RPTU**)

### Luxemburg

895
- Senior flood forecasting and data management expert at **AGE**
- General Director at **CGDIS**