# Peer review of "Comparing Flood Forecasting and Early Warning Systems in Transboundary River Basins"

_EGUsphere, 2025_

## Author Comment (AC1)

**Reviewer #2**

This paper addresses an important and timely topic on flood forecasting and early warning systems (FFEWS) in transboundary river basins, using the 2021 flood event as a key reference point. The paper contains a wealth of interesting insights from both literature and key informant interviews (KIIs), and the forensic perspective on the 2021 disaster is particularly valuable.

However, the overall flow and structure of the paper could be strengthened to help the main arguments and contributions emerge more clearly. In particular, the introduction and Section 3 would benefit from more explicit framing, stronger transitions between paragraphs, and a clearer delineation between preand post-2021 developments. The methodology also appears somewhat light and would benefit from revision to ensure there is a more systematic approach to data collection, content analysis and communication.

We want to thank you for your time and effort to review the manuscript. It is very nice to read that you think the results are interesting and valuable. We carefully responded to the issues you raised, which are outlined below. In our opinion, this led to a strong improvement of the manuscript. The section, figure and table numbers refer to the revised manuscript version.

**Major comments**

**1. Overall structure and flow**

The literature review provides valuable forensic insights from 2021, but the narrative currently mixes several issues in single paragraphs.
 More explicit structuring could help the key challenges and gaps stand out.

We carefully revised the introduction. We restructured the introduction into 3 different paragraphs: 1.1 A Flood Forecasting and Early Warning System (FFEWS), 1.2 Current challenges in FFEWS and 1.3 Research gap and aim. This required the revision and relocation of large pieces of text, including sometimes to other sections. Such movements and revisions include the part about the flood hazard zones, (which is now included in Section 4.4), the inclusion of additional information about the lack of uptake on IbF, and an expansion of the research gap.

• The introduction highlights important lessons but does not yet bring out the main research gap clearly.

Thank you for pointing this out. We did not realize that we did not clearly explain the research gap. Therefore, we now included the research gap in Section 1.3: "Given these challenges, it is crucial to compare and assess the current state of operational FFEWSs. Only very few studies assess operational FFEWSs in detail (e.g. Kaltenberger et al., 2020; Schroeter et al., 2021), and to the best of our knowledge none of them

assess all components of the FFEWSs chain, including communication and crisis management. Moreover, a transboundary analysis (e.g. on different warning levels in different countries) is lacking."

 The flow from paragraph to paragraph can be strengthened with more topic sentences and transitions that guide the reader through the logic. The introduction currently mixes a range of issues in one paragraph; separating them more clearly could help highlight the specific challenges the paper aims to address.

We have rewritten large parts of the manuscript to improve the structure and flow between the paragraphs. As an example, we added the following lines to improve the connection between different parts of the manuscript:

- Start of Section 3: "In this section, the key characteristics of FFEWS are presented based on recent literature and the expert interviews.
   First, we will present a general overview of the different systems (3.1), after which we will present the warning levels used (3.2), a detailed overview per country (3.3-3.6) and an overview of collaborations between the different regions (3.7)"
- o End of Section 3.7: "Below, we will outline the main challenges found in the study."
- Right before Section 4.1: "Even after those changes, the interviewees identified several issues that require attention to further improve FFEWSs (Table 2, bottom). We will further explain those challenges below."

As mentioned in response to your prevision comment, we also rewrote the Introduction, to improve the challenges identified and the key knowledge gaps. Moreover, we added sub-headings to all main sections (Section 1-5) of the manuscript to improve the readability.

 It remains unclear whether Section 3 is purely descriptive (based on literature) or includes empirical data from KIIs. Clarifying this distinction is essential.

Section 3 is based on a combination of both literature and information obtained in the expert interviews. To clarify this, we added the following to the start of section 3: "In this section, the key characteristics of FFEWS are presented based on recent literature and the expert interviews.".

**2. Definition of research gap and aim**

o The research gap and aim of the paper should be defined more clearly and earlier on.

We improved the definition of the research gap by including the following (Section 1.3):

"Given these challenges, it is crucial to compare and assess the current state of operational FFEWSs. Only very few studies asses operational FFEWSs in detail (e.g. Kaltenberger et al., 2020; Schroeter et al., 2021), and to the best of our knowledge none of them assess all components of the FFEWSs chain, including communication and crisis management. Moreover, a transboundary analysis (e.g. on different warning levels in different countries) is lacking."

We understand that you would prefer to have the research gap mentioned earlier. However, as NHESS is not a dedicated forecasting journal, we think it is important to first explain the core principles of FFEWS, before moving into the research aim. Although we did not move the research aim to an earlier section, the new structure (with sub-headings) allows readers to easily jump towards the research aim if they understand FFEWS already.

o The research question and focus area can also be stated more explicitly, ideally near the end of the introduction.

We mentioned the research questions in the last part of the introduction. However, we did not clearly state the focus areas of our research. Therefore, we included this at the end of the introduction: "We focus on the areas hit by the July 2021 flood, specifically The Netherlands (Limburg), the Rhineland-Palatinate (RLP) and North Rhine-Westphalia (NRW) federal states in Germany, Belgium (Flanders and Wallonia) and Luxembourg."

**3. Clarity of arguments in early sections (L83-L95)**

• From L83 onwards, the paragraph discusses communication issues but then shifts to examples where modelling outputs were inaccurate (e.g., flood zone delineation). These examples appear to relate more to forecast accuracy than to communication.

We mentioned the flood zone delineation issue to illustrate that people outside the official zones have a lower risk perception, and that therefore, warnings should be communicated differently to ensure that also these people take action. We understand that this was not clear. Therefore, we have rewritten the paragraph and restructured it to point out to the issues of 1) warning dissemination (i.e. not everyone received a warning, 2) communication issues (related to risk perception and behavior), and 3) cross-border collaboration and harmonization.

• The discussion around L90–L95 needs better alignment: the statement on flood awareness between in- and out-of-floodzone populations seems inconsistent with earlier points about fatalities outside the delineated zones, suggesting that flood extents exceeded forecasts.

With "flood zones" we referred to the legally defined flood zones in spatial planning policies, prior to the flood, and not to the forecasted flood (extent) itself. We understand that this was not clear, so we clarified it by changing "flood zone" to "the defined static flood hazard zones (as defined in spatial planning policies)". We hope that the paragraph in its current form is clearer.

o The sentence on adaptation motivation does not connect directly with the statement on flood warning access (L95).

With the changes mentioned above, we think this is clear now.

**4. Clarification of key terms and assumptions**

L165: Please elaborate on what constitutes a "clearly defined alarm level." When is this not clearly defined? It seems this may relate to
objective levels corresponding to forecast thresholds and expected impacts.

That's a good point. We now included that with clearly defined we mean "based on specific discharge or water level thresholds". We changed this in the new version of the manuscript.

**5. Section 3: Presentation and organization**

• The paragraphs describing the table and figure are difficult to follow. Consider adding more guiding sentences to help the reader navigate these visuals.

We rewrote the first paragraphs of Section 3 to a large extent, including more guiding sentences (see reply to comment 1 for some examples). We also restructured this section by dividing the first parts into two new sub-sections: 3.1 Overview of the operational FFEWS and 3.2 Warning levels in the region. As a response to reviewer #1, we also expanded Table 1. We included a detailed explanation of this table, in Section 3.1, to provide the reader with extra guidance.

**6. Depth and rigor of the methodology**

• The methodology section seems quite light: the sample size is small, and there is no indication of systematic coding or content analysis.

We performed a thorough literature review and combined this with expert interviews, which is a proven method to conduct assessment studies. However, the reviewer is right that the method description could be enhanced. This is why we expanded the methodology section (Section 2.2: Approach) by providing more information on the literature search (e.g. search engine and keywords used) and the number of articles/reports selected (16 scientific articles and 14 national reports). Of course, a much higher number of reports have been read before the selection took place. Furthermore, we interviewed 13 experts. Although this is indeed not a very large sample size, we took a number of measures to ensure that the results are valid. First, we ensured that we interviewed at least 1 forecaster and 1 person directly involved in

the crisis management in every country. This is added to the manuscript. Second, the interviews were very productive and lasted long (mostly >2 hours), which ensured that the information was interpreted correctly. Third, almost all interviewees reviewed the paper before submission to validate the results and the co-authors themselves are also experts across all study regions. Therefore, we are convinced that the gathered results reflect reality and provide a holistic view on the current state of FFEWS. We analysed the content based on a thematic analysis, conducted in every pillar of the early-warning chain. We describe this in the middle of the section.

o For research question (b), a more in-depth analysis of communication materials would strengthen the conclusions.

In Section 4.4, we used 15 different sources to show why communication challenges exist (e.g. missing communication protocols, lack of actionable information, fragmented pieces of information from multiple stakeholders). We have rewritten this section, and we think that in the current form it gives a clear view on the challenges and how those can be addressed. Furthermore, we added a clear recommendation about improving communication (Section 5.2): "Forecast communication: Implement a structural evaluation of warning communication chains, to ensure that warnings are consistently communicated and correctly interpreted between different organizations, and to the local public, and ensure that the information is tailored to their needs."

**7. Treatment of transboundary dynamics**

o The transboundary challenges could be brought out more clearly, particularly regarding data sharing and alignment of alert levels.

Because of your comment, we realized that we did not focus enough on the transboundary aspects of data sharing. Therefore, we expanded the paragraph on transboundary data sharing and now list the most important transboundary commissions which facilitate data exchange (Section 3.7). Our interviewees actually did not mention many challenges in transboundary data sharing but rather emphasized that this is already quite well organized. Therefore, we moved this section from Section 4 ("Key challenges remaining after the 2021 flood") to Section 3 ("Key characteristics of the operational FFEWS").

We also revised Section 4.1, mainly to emphasize that the differences in alert levels are not only present within different regions of a country (and between the meteorological and hydrological agencies), but also between the different countries.

o Consider including another figure to illustrate the communication side of these systems.

This is a very nice idea, but we think this can only be included in a proper way in a follow-up study. The manuscript is already quite long (especially with the new Table 2) and therefore we think an extra figure (including explanation) better suits a follow-up study dedicated to forecast communication.

**8. Evaluation of effectiveness**

o The paper provides a rich description but limited critical evaluation of the accuracy and effectiveness of the new developments in FFEWS.

In Section 4, we provide a detailed description of the challenges that remain after the upgrades triggered by the 2021 flood event. In the old manuscript, it was maybe not clear that this section addressed the remaining issues after 2021. Therefore, we renamed the section to "4. Key challenges remaining after the 2021 flood". Moreover, some parts of the section have been expanded. We cannot really provide clear conclusions yet on the effectiveness of the new developments after 2021, as we will need to be careful with that and the only way you can really know this is after a large flood happened in the area.

o It would be useful to reflect on whether improvements have been validated and to synthesize recommendations based on that assessment.

We synthesized the issues remaining after the 2021 flood in four key challenges. We now mention these challenges clearly in Table 2. We also developed four concrete recommendations to address those challenges, one for every challenge. These recommendations are listed at the end of the manuscript, in Section 5.2:

- 1) Streamlined warning levels: investigate whether harmonization of flood warning levels and color codes between different countries and within different regions of a country, improves communication and decision-making.
- 2) Warnings for catastrophic events: Assess the added value of more extreme warning levels, such as the (dark) purple levels in Luxembourg and some German regions.
- 3) Impact-based forecasting: accelerate the development of operational impact-based forecasting systems. This in turn requires an expansion of resources and knowledge on impact-based forecasting, and a stronger collaboration across sectors, such as between forecasting agencies and emergency management.
- 4) Forecast communication: Implement a structural evaluation of warning communication chains, to ensure that warnings are consistently communicated and correctly interpreted between different organizations, and to the local public, and ensure that the information is tailored to their needs.

In line with our response above, it is good to mention that a validation of the improvements can only reliably be done after a flood event took place.

**9. Scope and temporal framing**

• The content currently mixes pre- and post-2021 developments, leading to ambiguity about the study's temporal focus. Clarify whether the analysis primarily concerns pre-2021 systems, post-2021 changes, or both.

Thank you for this very useful comment. We totally agree that this was a weak spot of our previous manuscript version. Therefore, we took several measures to improve the definition of the study's temporal focus. First, we renamed Section 4 to "Key challenges remaining after the 2021 flood" to stress that those are the challenges remaining after the improvements triggered by the 2021 flood. Secondly, we included a new table (Table 2, see below) to list the major changes of the FFEWS triggered by the 2021 flood in the different regions. We also describe all those changes in the beginning of Section 4.

**Minor Comments**

L1, L119: Write out abbreviations such as "bn" and "mm."

Thank you. This is done.

• L83-L95: Improve alignment between discussion of communication issues and modelling inaccuracies.

Done (see comments above on point 3)

L165: Clarify "clearly defined alarm level."

Here we meant "based on specific discharge or water level thresholds" (see response on point 4)

• L492: Clarify the source of the quoted text.

The quoted text originates from the interviews we had in Wallonia. We changed it to: "Our interviewees in the Belgium region Wallonia stated that "...there should be a new discussion about the need for an extra level of warning (e.g., dark red) for really extreme situations"

• L495: The phrase "cope for decision makers" is awkward—consider revising to "decision-making challenges" or similar.

Done

• Throughout: Review paragraph transitions and ensure topic sentences clearly indicate the purpose of each paragraph.

As you can see in our track changes document, we thoroughly revised the entire manuscript, including new paragraph transitions, a better structure and improved writing style.

Table 2. Most important changes of the Flood Forecasting and Early Warning Systems (FFEWS) triggered as response to the 2021 floods, and the 4 key challenges remaining in different European regions and countries.

| Country                                | Germany
(RLP)                                                                                                                                                                                                                  | Germany
(NRW)                                                | Luxembourg                                                                                 | The Netherlands                                                                                                       | Belgium
(Flanders)                                                                                        | Belgium
(Wallonia)                                                                                   |
|----------------------------------------|-----------------------------------------------------------------------------------------------------------------------------------------------------------------------------------------------------------------------------------|-----------------------------------------------------------------|--------------------------------------------------------------------------------------------|-----------------------------------------------------------------------------------------------------------------------|--------------------------------------------------------------------------------------------------------------|---------------------------------------------------------------------------------------------------------|
| Major changes since the 2021 floods    | DE-alert system, and
addition of extra
warning level for very
extreme floods (dark
purple)                                                                                                                            | DE-alert system, and initiation of a new flood forecasting team | LU-alert system,
including a new
warning level for
"immediate danger"
(purple) | A new multi-year
programme on flood
safety in Limburg
(WRL) and updated
flood disaster
management plan | Optimized hydrological
warning thresholds (see Fig. 3)
and revised Emergency and
Intervention Plans | Flood expertise units (CELEX), updated flood portal and move from deterministic to ensemble forecasting |
| Key challenges remaining
after 2021 | Streamlined warning levels and thresholds  Different warning levels, thresholds and color codes are used in the between and within different countries and organizations.                                                         |                                                                 |                                                                                            |                                                                                                                       |                                                                                                              |                                                                                                         |
|                                        | Warnings for catastrophic events  The highest thresholds are often too conservative (too low) and thus do not reflect major societal flood impacts.                                                                               |                                                                 |                                                                                            |                                                                                                                       |                                                                                                              |                                                                                                         |
|                                        | Impact-based forecasting  While impact-based forecasting is recognized as crucial to make forecast-informed decisions, such systems are rarely operationally implemented.                                                         |                                                                 |                                                                                            |                                                                                                                       |                                                                                                              |                                                                                                         |
|                                        | Communication before and during the crisis phase  Effective communication between different organizations (e.g. between meteorological agencies, hydrological agencies, crisis management authorities) and to civilians remains a |                                                                 |                                                                                            |                                                                                                                       |                                                                                                              |                                                                                                         |
|                                        | challenge.                                                                                                                                                                                                                        |                                                                 |                                                                                            |                                                                                                                       |                                                                                                              |                                                                                                         |